# Designing for Shape Memory in Additive Manufacturing of Cu–Al–Ni Shape Memory Alloy Processed by Laser Powder Bed Fusion

**DOI:** 10.3390/ma15186284

**Published:** 2022-09-09

**Authors:** Mikel Pérez-Cerrato, Itziar Fraile, José Fernando Gómez-Cortés, Ernesto Urionabarrenetxea, Isabel Ruiz-Larrea, Iban González, María Luisa Nó, Nerea Burgos, Jose M. San Juan

**Affiliations:** 1Departamento de Física, Facultad de Ciencia y Tecnología, Universidad del País Vasco (UPV/EHU), Apdo. 644, 48080 Bilbao, Spain; 2CEIT-Basque Research & Technology Alliance (BRTA), Manuel Lardizabal 15, 20018 Donostia-San Sebastián, Spain; 3Universidad de Navarra, Tecnum, Manuel Lardizabal 13, 20018 Donostia-San Sebastián, Spain; 4LEITAT, c/de la Innovació 2, 08225 Terrasa, Spain

**Keywords:** additive manufacturing, laser powder bed fusion, shape memory alloys, Cu–Al–Ni, martensitic transformation

## Abstract

Shape memory alloys (SMAs) are functional materials that are being applied in practically all industries, from aerospace to biomedical sectors, and at present the scientific and technologic communities are looking to gain the advantages offered by the new processing technologies of additive manufacturing (AM). However, the use of AM to produce functional materials, like SMAs, constitutes a real challenge due to the particularly well controlled microstructure required to exhibit the functional property of shape memory. In the present work, the design of the complete AM processing route, from powder atomization to laser powder bed fusion for AM and hot isostatic pressing (HIP), is approached for Cu–Al–Ni SMAs. The microstructure of the different processing states is characterized in relationship with the processing parameters. The thermal martensitic transformation, responsible for the functional properties, is analyzed in a comparative way for each one of the different processed samples. The present results demonstrate that a final post–processing thermal treatment to control the microstructure is crucial to obtain the expected functional properties. Finally, it is demonstrated that using the designed processing route of laser powder bed fusion followed by a post–processing HIP and a final specific thermal treatment, a satisfactory shape memory behavior can be obtained in Cu–Al–Ni SMAs, paving the road for further applications.

## 1. Introduction

Additive manufacturing (AM) constitutes the new paradigm of the millennium for materials processing, opening possibilities for final–shape technology production [1], not only for polymers but also for metallic materials [2,3,4]. Metals and alloys require a precise control of the microstructure to guarantee their mechanical properties, and a huge international effort is being devoted to the optimization of AM parameters to produce many structural alloys, from steels or superalloys [3,4] to new emerging materials such as high entropy alloys [5,6] or TiAl intermetallic alloys [7]. Most of the additive manufactured metallic materials are produced by powder bed fusion techniques such as laser powder bed fusion (LPBF) or electron beam melting (EBM), as well as, for instance, through flow–based deposition techniques (LMDs) or wire arc melting (WAAM), with a variety of methods that are described in the literature [1,2,3,4]. One of the challenges of these AM techniques is to produce functional materials with active properties, which require a particularly compact and reproducible final microstructure. Among them, shape memory alloys (SMAs), exhibiting shape memory and superelastic properties, based on a reversible thermo–elastic martensitic transformation (MT) [8,9], are attracting scientific and technological attention from the AM community. Initially, the effort was focused on the AM of Ti–Ni SMAs [2] due to its huge number of applications [10], including aerospace ones [11], as well as, in recent years, outstanding progresses being made to optimize the processing parameters of several alloys based on the Ti–Ni system [12,13,14]. Some overviews on the AM of SMAs were recently published [15,16,17], evidencing that although Ti–Ni is the most worked SMA, other kinds of SMAs are attracting attention, specifically Cu–based SMAs [18,19,20,21] and magnetic SMAs [22,23]. In particular, SMAs based on the Cu–Al system are compelling from a technological point of view, due to their good thermomechanical properties as bulk materials, even as high–temperature SMAs [24], and due to their exhibiting of good shape memory effects at the nanoscale [25], which are being applied in several sectors as, for instance, the aerospace industry. However, they are challenging for AM processing due to the inherent difficulties to process copper and copper–rich alloys, associated with their high thermal conductivity and the high reflectance of the red and infrared lasers [26]. In previous works, the powder metallurgy (PM) route of these SMAs produced materials with good shape memory and superelastic behavior [27,28,29]. Regarding the PM of SMAs, it must be remarked that the production of Cu–based SMA powders is much easier and less expensive than the equivalent Ti–Ni SMA powders. The PM route also allows for the development of Cu–Al–based SMAs by severe plastic deformation with very fine microstructures [30], and at present AM is the base of many processes opening the way to obtain near–net–shape components of SMAs with unforeseen technological capabilities.

Nevertheless, the works in the literature devoted to the AM processing of Cu–based SMAs have not yet achieved good thermal martensitic transformation nor good shape memory behavior [31]. This could be attributed to a degradation of the microstructure during the AM processing, due to stable phases precipitation, preventing the presence of a sample fully in the austenite phase, as required for the further martensitic transformation. As it will be showed, a thermal treatment for the functionalization of the LPBF sample will be necessary. Then, in the present work, the complete methodology of the AM processing of Cu–Al–Ni SMAs is approached, from powders production to laser powder bed fusion fabrication, followed by some post–processing thermal treatments. The microstructure of the sample and the thermal transformation are fully characterized. The final additive manufactured alloy exhibits an excellent thermal martensitic transformation as well as good shape memory behavior.

## 2. Materials and Methods

### 2.1. Selected Alloy and Powder Preparation

Gas atomization is one of the leading methods to manufacture the powders used by the PM industry. It is attractive due to its applicability to many alloys, reasonable process control, high productivity, and the possibility to produce spherical particles with good packing characteristics, which are the ones used in AM. The resulting particle size depends on several operational variables, geometrical parameters, and physicochemical properties of gas and melt [32,33,34].

In the present work, the powders of the SMAs were obtained in the atomizer of the CEIT, which is a small–scale research atomization unit PSI model HERMIGA 75/3VI (Phoenix Scientific Industries Ltd., Hailsham, UK). A description of the equipment is conducted in [35]. The Cu–Al–Ni alloy for LPBF–fabricated samples was chosen with a nominal composition of Cu_82.5_Al_13.5_Ni_4_ (wt.%) to obtain transformation temperatures above room temperature [36]. The Cu–Al–Ni alloy was produced using high purity elements and adding the Ni as a master Cu–Ni alloy in order to reduce its melting point and to avoid the losses of Al by evaporation or oxidation during the melting process. Melting was performed under a high purity Ar atmosphere and the atomization gas was also Ar at a pressure of 55 to 60 bars. An overheating of around 200 °C was used above the melt temperature, which in the furnace corresponded to 1087 °C. As the amount of powder produced per atomization is about 3 kg, two atomizations were necessary to obtain the quantity of powders required for the LPBF additive process. For processing by LPBF, only the fraction of spherical powders below 63 microns in diameter should be used, and after sieving only about 70% of powders are retained per atomization (2.2 kg). The particle size distribution of the powders was measured by laser diffraction in a Sympatec Helos (H0852) equipment (Sympatec GmbH, Clausthal-Zellerfeld Germany), and the cumulative and frequency curves are shown in Figure 1.

The chemical composition of the two atomized powders was measured by inductive coupled plasma spectroscopy (ICP) in a Varian 725–ES ICP–OES equipment (Agilent Technologies, Santa Clara (CA), USA). The oxygen content was measured using an instrument LECO TC–400 and a LECO CS–200 (LECO Corporation, St. Joseph (MI), USA), for carbon and sulfur. The results are remarkably homogeneous, as seen in Table 1, where the low content of O and C can be noticed and the N and S appear as residual impurities. It must be noted that the chemical homogeneity of the alloy is fundamental because the martensitic transformation temperatures in SMAs are very sensitive to the chemical composition of some of their elements, for instance to Ni in Ti–Ni [37] and to Al in Cu–Al–Ni [38] SMAs. Once the compositions of the powders from both atomizations were verified to be similar, all powders were mixed for further AM processing.

The morphology and appearance of the powders were characterized by the secondary electron signal (SE) of a Schottky scanning electron microscope (SEM), JEOL JSM–7000F (JEOL Ltd., Tokyo, Japan) equipped with back–scattered (BSE), X–ray (EDX, Oxford, UK), and electron back–scattered diffraction (EBSD, HKL) detectors (both from Oxford Instruments plc., Abingdon, UK). A couple of images of powders with two different sizes (<63 μm) are presented in Figure 2, to show the spherical shape of particles.

As the powders are fast quenched during atomization, they exhibit the MT of the SMAs during cooling, being in martensite at room temperature, as shown in the SEM–BSE image of Figure 3a, where the martensite variants are clearly seen at the surface of the powder particle. The thermal transformation of the powders was measured by differential scanning calorimetry (DSC) in a DSC TA Q2000 (TA Instruments, New Castle (DE), USA), with He atmosphere in the measure cell, in the temperature range of −100 to 150 °C at a cooling–heating rate of 10 °C/min. In Figure 3b, the DSC thermograms of the powders illustrate the direct MT (during cooling) and the reverse MT (during heating), as well as the normalized transformed mass fraction n(T), obtained by integration of the DSC thermograms, as presented in Figure 3c.

From the n(T) curves, the MT temperatures, Ms (martensite start), and Mf (martensite finish) during cooling, as well as As (austenite start) and Af (austenite finish) during heating and the thermal hysteresis, can be easily obtained. It is a common practice [38] to determine the MT temperatures at 2% and 98% of the transformation, so at 0.02 and 0.98 of the transformed mass fraction n(T), and the transformation temperatures for the powders (<63 μm in diameter) are: Ms = 47.5 °C, Mf = −8.3 °C, As = 25.9 °C and Af = 91.6 °C, and the thermal hysteresis, measured at n(T) = 0.5, is ΔT = 26.4 °C.

### 2.2. LPBF Parameters and Scanning Strategy

The LPBF samples studied in this work were processed in a Renishaw AM400 LPBF machine (Renishaw plc., New Mills, UK), with the help of the so–called reduced build volume (RBV) module, which allows for the construction of AM samples with the use of small amounts of metal powder (in the order of 2 to 4 kg). Cubic samples of 10 × 10 × 10 mm^3^, above the small pyramids contacting with the base of the RBV module, were fabricated using different sets of parameters. Figure 4a illustrates the RBV platform with the simulated samples, and Figure 4b is an example of the as–LPBF samples.

A series of samples with different experimental conditions were obtained varying the scanning spacing, the hatch distance *d_h_*, the scanning speed *ν*, and the laser power *P*, because from the expression of the volumetric energy [39], the energy per surface area *E* (J/mm^2^*)* can be expressed as a function of these three values and is given by Equation (1):(1)$$E=\frac{P}{d_{h}\cdot v}$$

As the scanning strategy also has a great influence on porosity and the final consolidation of the samples, two different scanning patterns were used during the LPBF process; total fill (TF), which consists on a spire–like movement of the laser, Figure 5a, and meander (M), which consists of a 90° rotation of the scanning direction to cover the complete layer as seen in Figure 5d. In Figure 5, the scanning strategy (a,d), a general view of the grown samples (b,e), and the corresponding optical micrographs of the polished samples (c,f), obtained with a Leica–Reichert MET3 MF4 A/M (Leica Microsystems GmbH, Wetzlar, Geremany), are presented as representative examples. 

Two scan passes were applied: in the case of sample SM–1, a first low energy scan (0.04 J/mm^2^) was used as pre–heating, whereas in other cases, such as samples SM–2 as well as HIP–1 and HIP–2, a second low–energy scan was used for stress releasing, such as is presented in Table 2, which summarizes the complete set of parameters and scan conditions used in the present work.

### 2.3. Post–Processing Treatments

In the literature [20,21,31], some LPBF processing and measurements were made on the basis that the LPBF processing has associated a very fast cooling of the material and does not require any further post–processing treatments. However, it is important to indicate that polycrystalline Cu–Al–Ni SMAs exhibit a strong tendency to precipitate at the grain boundaries [40]. The presence of precipitates alters the performance of the alloy and could even render it useless. For this reason, we found that a post–thermal treatment is necessary in Cu–Al–Ni SMAs, and this aspect will be discussed in the following sections.

After the LPBF processing, different thermal treatments were conducted on the LPBF–built samples to test their viability. Both of them start with a solid solution treatment at 900 °C, 30 min in argon atmosphere, ensuring that the β phase, called austenite, would be well solubilized. Then, through a fast enough cooling, in order to avoid the precipitation of the stable phases, this β phase can be frozen in metastable state at lower temperatures for its further transformation in martensite. However, due to the presence of porosity in the samples, as can be observed in Figure 5c,f, there was a worry that a very fast cooling could produce inter–granular fractures on the samples, so some of them were quenched on boiling water (slow quench), while others were quenched in iced water (fast quench, which is the usual method for the functionalization of Cu–Al–Ni SMA single crystals [24,41]). The samples quenched at 0 °C are then aged for 24 h at 180 °C to accomplish the L2_1_ atomic order of the metastable β phase [42] and stabilize the transformation temperatures that otherwise evolve with cycling [43,44].

Finally, to reduce the porosity observed in the as–built LPBF processed samples, as seen in Figure 5c,f, a hot isostatic pressure (HIP) process was conducted on some of the LPBF samples, named HIP–1 and HIP–2 in Table 2. The conditions for HIP were 875 °C and 140 MPa during 3 h in a ASEA (ABB) QIH–6 equipment (ABB Corporation, Zurich, Switzerland). To track the porosity of the samples, their microstructures were characterized by optical microscopy (Leica DMRXA 2) with Nomarski interferential contrast, and after that by SEM. Samples were encapsulated in epoxy resin and mechanically grinded and polished down to a particle size of 1 μm. Finally, samples for SEM and EBSD were electrochemically polished with a solution of HNO_3_ at 33% in methanol at T = 0 °C, with an applied voltage V = 10 V for 5 s, in a Struers LectroPol–5 (Struers, Cleveland, OH, USA).

## 3. Results

### 3.1. As–LPBF Processed Samples

Single melt scans with pre–heating (SM–1) and post–heating (SM–2) were initially used. Then, double melt scans, with different surface energy and scan strategy, were used to process samples DM–1 to DM–4. Among the two scan strategies shown in Figure 5, it was observed that the TF strategy has a tendency to produce alignment of the pores in the diagonal of the square piece, and consequently it is not recommended for melting, but only eventually for pre– or post–heating.

One of the main challenges when building LPBF samples is choosing the optimal parameters to minimize the porosity of the final material. It was observed that fast scanning speeds, large hatch distance, and/or low laser power, do not provide enough energy for all the material to melt. Apart from pores caused by trapped gas bubbles, the low energy provided to the material can result in non–melted powder that forms occlusions inside the pieces which is an important cause for porosity too. This porosity is observed in samples SM–1 and SM–2 as well as in samples DM–1 to DM–4, and some examples are shown in the SEM–SE micrographs of Figure 6.

Another crucial aspect in SMA is to determine whether a further functionalization treatment is needed or not, and the effect that different thermal treatments have over the LPBF–produced pieces. In Figure 7a, a fine distribution of nanometric precipitates is observed by SEM–SE throughout the LPBF DM–3 sample; this small precipitate can also be seen in Figure 6b taken in a region of the SM–1 sample. 

In Figure 7b, taken in a similar sample, another bigger precipitate was identified by EDX as the γ equilibrium phase. Even if the cooling process is fast enough at the melting surface to avoid the nucleation and further growth of the precipitates, the further melt of new layers will heat the layers beneath it, producing the nucleation and growing of precipitates, and this effect should not be neglected. Even if the precipitates are small, as seen in Figure 7, their presence alters the composition of the matrix, therefore changing the transformation temperatures and lowering the mechanical and functional properties of the alloy. For the particular composition of the alloy presented in this work, which is placed in the hypereutectoid region [45], the pro–eutectoid precipitates must correspond to the stable Al–rich γ phase [40] shown in Figure 7b. However, the cuboidal nanometric precipitates are Cu–rich with a high amount of oxygen, about 9% as measured by EDX, and can be identified with the α phase, which has higher oxygen solubility than the β phase, and the premature apparition of these nanometric α precipitates could be attributed to the excess of oxygen. The original alloyed powders are oxidized at the surface and the oxygen is incorporated to the pool melt during the laser fusion; during solidification, the high temperature β phase is saturated in oxygen, and during cooling it segregates it through the precipitation of nanometric precipitates rich in oxygen. However, these Cu–rich precipitates do not have the expected equilibrium stoichiometry. A deep microstructural study of these precipitation mechanisms is in progress, but it is out of the scope of the present work.

### 3.2. Post–Processing: Thermal Treatments

Once the precipitates are formed, the only way to restore the microstructure is through a solid solution treatment, which is performed by heating the material up to 900 °C, in an inert Ar atmosphere, for 30 min and then quenched. Some samples were quenched in boiling water, (TT–100), while others were quenched in iced water, and then underwent an aging of 24 h at 180 °C, (TT–0), to stabilize the atomic order of the alloy and consequently make the transformation temperatures stable [44]. An example of the influence of both thermal treatments is shown in Figure 8, where SEM images are comparatively presented for the LPBF–processed sample SM–1, after the TT–100 treatment in Figure 8a (SEM–SE) and after the TT–0 treatment in Figure 8b (SEM–BSE). These images clearly illustrate that even a quench at 100 °C is not fast enough to avoid the precipitation of the stable phases, which are well revealed in Figure 8a, whereas the sample quenched at 0 °C and aged at 180 °C is free of precipitates, as seen in Figure 8b.

As it was expected, the presence of precipitates has a noticeable influence on the martensitic transformation evidenced in the DSC thermograms shown in Figure 9a, and in the transformed fraction curves shown in Figure 9b. Indeed, in Figure 9, the martensitic transformation of the LPBF–processed samples SM–2 is presented for the as–LPBF state and after the thermal treatments TT–100 and TT–0, in comparison with the corresponding curves obtained for the atomized powders. During the as–built process or even during a slow quench (TT–100), a small amount of Al–rich phases precipitate (Figure 7b or Figure 8a) and the austenite matrix are slightly depleted in aluminum, and consequently the MT temperatures should shift into higher temperatures [24,36,38], as it is shown by the violet and red curves in Figure 9, corresponding to the as–built and TT–100 samples, respectively. On the contrary, with the fast quench (TT–0) for which the sample is precipitates free, corresponding to the blue curves in Figure 9, the matrix exhibits the stable MT behavior (Ms = 76.4 °C, Mf = 23.5 °C, As = 65.2 °C, and Af = 90.7 °C) with a slightly narrower hysteresis of ΔT = 17.2 °C. In this case, the complete MT is also shifted to higher temperatures with respect to the one exhibited by the atomized powder. This last aspect is a general observation that will be discussed in the next section.

### 3.3. Post–Processing: HIP

Finally, in an attempt to reduce the present porosity, a process of HIP was conducted in a new set of LPBF samples. The conditions were varied in correspondence with the previously observed results. On the one hand, the overall energy per surface area was increased (see HIP–1 and HIP–2 in Table 2) in order to reduce the amount of non–melted material, which was responsible for the kind of porosity shown in Figure 6; the laser power was slightly increased, a smaller hatch distance was used, and low scanning speeds were applied. On the other hand, an HIP treatment will be able to close the small pores associated with thermal contraction after melting. This way, a better compaction of the material should be expected from the HIP–treated LPBF pieces. It is important to mention that the HIP–1 and HIP–2 samples were thermally treated after the HIP. According to the results presented in Figure 8 and Figure 9, in this case the chosen treatment was the one preventing precipitation, namely 900 °C for 30 min, followed by a quench in iced water and an aging of 24 h at 180 °C (TT–0). Figure 10 presents the interferential contrast optical micrographs of the LPBF–produced material after the HIP (HIP–1 in Table 1) and thermal treatment TT–0; practically no porosity and a complete martensitic transformation are observed throughout the sample.

Regarding what concerns the thermal characterization of the HIP–treated samples by DSC, Figure 11a shows the results for the as–built SM–2 sample (blue curves) in comparison with the ones after the HIP process (red curves); both samples are TT–0 treated. The results show that the HIP process does not produce any significant changes over the value of the transformation temperatures, with respect the LPBF single melt sample without HIP.

The curves of the transformed fraction presented in Figure 11b show that in the HIP sample, the MT is only slightly shifted (Ms = 79.9 °C, Mf = 28.1 °C, As = 68.4 °C, and Af = 92.6 °C) and the hysteresis, measured at n(T) = 0.5, remains rather small, as ΔT = 16.6 °C after HIP. This point will be also discussed in the next section.

## 4. Discussion

In Cu–Al–Ni SMAs, the MT can take place through two different martensites [46] whose crystallographic parameters were identified in the literature by neutron diffraction for the monoclinic $\beta_{3}^{\prime }$ martensite [47] (spatial group C2/m, a = 1.38017 nm, b = 0.52856 nm, c = 0.43987 nm, β = 113.6°) and by X–ray synchrotron diffraction for the orthorhombic $\gamma_{3}^{\prime }$ martensite [48] (spatial group Pmmn, a = 0.53424 nm, b = 0.42244 nm, c = 0.43896 nm). The orientation relationships of both martensites were determined by in situ TEM [49], and the sub–index 3 refers to the atomic order L2_1_ of the original *β* phase, according to the proposed nomenclature for martensites [50]. With these parameters, the martensite appearing in the present LPBF samples, in Figure 8 and Figure 10, can be identified through the SEM–EBSD patterns, which were performed at 20 KV and at 185 mm distance. The SEM–BSE micrograph of Figure 12a shows a self–accommodating group of martensite variants in sample SM–2, and Figure 12b shows, as an example, the EBSD pattern corresponding to the point marked in Figure 12a. The indexation of patterns was performed using the HKL Flamenco software and corresponds to the monoclinic $\beta_{3}^{\prime }$ martensite in all the measured points with an orientation relative to the surface of the sample given by Figure 12d, where the atomic structure of the martensite lattice is showed for the EBSD pattern shown in Figure 12b and indexed in Figure 12c. This study was performed in different martensites along the samples, and in all cases the EBSD–indexed patterns correspond to the monoclinic $\beta_{3}^{\prime }$ martensite. This martensitic phase was expected for the selected alloy composition, which is placed on the right side of the eutectoid of the Cu–Al–Ni phase diagram [51,52], but in the case of the Ni–rich concentration range corresponding to the $\beta_{3}^{\prime }$ martensite [36,38]. Indeed, this result also agrees with the DSC results, showed in Figure 9 and Figure 11, because the $\beta_{3}^{\prime }$ phase has lower hysteresis than the $\gamma_{3}^{\prime }$ martensite.

The use of the same alloy and powders for all the fabricated samples allows for the analysis of the LPBF samples processed under different conditions, whose final aspect and properties are vastly different. These results point to the great importance that the fabrication parameters and the further thermal treatments have over the properties of Cu–Al–Ni SMAs processed by LPBF. The amount of energy provided to the powder affects the porosity and overall consolidation of the pieces, in agreement with previous reported works on Cu–based SMAs processed by LPBF [18,19,20,21,31]. From the study presented in Figure 5, the total fill scan strategy should not be used in order to minimize the generation of pores. The double melt scan could give good results, but the obtained porosity is very sensitive to the laser power [20] and a careful analysis of this parameter is out of the scope of the present paper. Unfortunately, the scanning strategy selected for the second melting tests was not the right one, exhibiting a rather high porosity. However, some interesting information could be retained from these experiments, as it will be further discussed. An increase in the local surface power was necessary to avoid the lack of fusion and as the laser power was increased, a smaller hatch distance was used and smaller scan rates were employed; new parameters were applied for the LPBF process in samples HIP–1 and HIP–2, as shown in Table 1. In addition, an HIP of the as–grown LPBF samples was revealed to be very useful to decrease the porosity, practically suppressing the small pores associated with local contraction during the solidification process, as shown in Figure 10.

The presented results evidence that appropriate post–processing thermal treatments are crucial to obtain a reproducible martensitic transformation and good final functional properties, and this aspect is particularly important in the case of Cu–based SMAs. In Figure 13, the thermal transformation of the samples is summarized for the different processing conditions. First, it is important to remember that the Al content in this SMA reduces the transformation temperatures around 170 °C per 1 wt.% of Al [24,36,38], so even a small number of precipitates might significantly alter the properties of the transformation. Indeed, the generation of precipitates in the hypereutectoid region depletes the Al composition of the matrix, giving place to an increase in the transformation temperatures, as well as an enlargement of the transformation range due to the gradients in Al concentration created around the precipitates. Even though the laser heats a very small region at a time and the cooling rate of the material in that area is very high, the heat produced by the laser must be somehow dissipated and may be absorbed by the material surrounding and directly beneath the melted area, due to the high thermal conductivity of copper. Then, the precipitation mechanisms of the stable phases may be triggered in the layers beneath the melting area.

This phenomenon explains why some precipitation is present in the as–grown alloys, with no solid solution treatments, as seen in Figure 7, and why the precipitates are very small in size, as they are not given enough time to grow and coalesce. The thermal MT in the as–processed sample, represented by the violet curve in Figure 13, is shifted to higher temperatures and enlarged due to the depletion in Al and the local gradients of the matrix, in contrast to the sample that was further treated with the fast–cooling TT–0 treatment, represented by the blue curve in Figure 13, in which the sample is precipitation–free and the MT is observed at a lower temperature, as well as being narrower than in the case of the as–grown sample, due to the suppression of the local concentration gradients. The same scenario, or even worse, takes place when the sample undergoes a too–slow quench (TT–100) from the austenite, allowing for the precipitation of the stable phases, as seen in the red curve in Figure 13. The heat flow peaks measured by DSC exhibit some asymmetry that can be explained as follows. In the low temperature side, the peak becomes enlarged because the martensite plates must nucleate in an environment with increasing internal stresses as the cooling progresses. A slight asymmetry is also observed in the high temperature side for the samples SM–2 as LPBF and SM–2 TT–100, which can be associated with the precipitation of the stable phases. The stable γ precipitates not only produce a depletion of aluminum in the matrix, but also generate some local compositional gradient, which is responsible for a distribution of the transformation temperatures.

At this stage, it could be asked: Why do all processed samples exhibit higher transformation temperatures than the original powders? This is a general observation on Cu–based SMAs produced by powder metallurgy [27,28,29], and the explanation is associated with two different aspects: the precipitation of the stable phases [40] and the influence of the internal stresses on the martensitic transformation. The powder particles solidify in a very relaxed condition, even with a high cooling rate, due to the large free surface with respect to the volume, and consequently the MT takes place in a low–stress condition. This means that the MT is dominated by the thermal driving force, taking place at lower temperatures. Each particle of powder is coming from a particular point of the melt before the atomization, and may transform at a precise temperature because they have a homogeneous chemical composition, confirmed by EDX analysis at the SEM. However, the melt before atomization may exhibit a slight chemical concentration gradient, which is translated into the powders since in the end all powders are mixed, and hence when measuring by DSC a distribution of the MT temperatures is expected. Then, the MT of the powders, represented by the green curve in Figure 13, is significantly broadened. On the contrary, during the LPBF processing, the melt pool will produce a statistical homogenization of the local concentration of the melted powder, so the MT becomes narrower than in the powders. In addition, the fast solidification, constrained by the surrounding solid, will generate a high level of internal stresses in the austenite, and the MT during cooling will be stress–assisted, taking place at higher temperatures. In Table 3, the martensitic transformation temperatures, thermal hysteresis, and transformation enthalpies for all the processed samples described along the present work, and presented in Figure 13, are summarized. The variation of the transformation temperatures extracted from the curves of Figure 13 evidences an influence of the microstructure on the MT in agreement with the above comments. 

However, it is worth noting that the measured enthalpies of transformation, ΔH, are practically constant except for the powders that show a slight decrease, in agreement with the previous experimental results and with the models predicting the dependence of the enthalpy with the transformation temperature [24,53]. This means that the precipitates observed in Figure 6, Figure 7 and Figure 8, and produced during the LPBF and even during further slow quenching treatments, are in a mass fraction that is too small to noticeably modify the transformation enthalpies; in any case, the highest enthalpy is obtained for the HIP–2 TT–0 sample, which is precipitates free. Nevertheless, the main precipitates are expected to be of the gamma equilibrium phases [40,45], which are richer in Al than the austenite phase and consequently their precipitation makes the austenite slightly poorer in Al, hence with higher MT temperatures due to their high sensitivity to the Al [38].

The influence of the internal stresses and their relaxation during the MT in Cu–Al–Ni was quantified through adiabatic calorimetry and neutron diffraction [54,55,56] and is in agreement with the above explanation. Moreover, this concept makes it possible to explain why in the case of the double melted samples (TT–0), represented by the orange curve in Figure 13, the MT is placed in between the as–LPBF processed by a single melt sample (TT–0) and the one from the powders. The low energy double melt may produce a noticeable relaxation of the local stresses accumulated during the first melt, as is evidenced by the increase in the grain size as well as by the increase in the size of martensite variants [20]. Then, the relaxation of the local internal stresses may be responsible for the shift in temperature of the MT in double melted samples. Finally, it is worth noting that the HIP–2 (TT–0) sample, represented by the black curve in Figure 13, exhibits a similar thermal behavior to the as–processed and treated sample, SM–2 (TT–0), with only minor variations regarding the temperatures and the hysteresis.

The Cu–Al–Ni SMAs processed by LPBF, and further HIP–ed with a TT–0 treatment, exhibit superelastic and shape memory effects. Indeed, two series of compression tests were performed on this sample, at several temperatures and for different maximum strains. The tests were made in an Instron 4467 machine equipped with a heating chamber; the strain was measured with an extensometer Instron of 10 mm gauge length and the temperature was measured with a thermocouple in contact with the sample. The testing temperatures were chosen to be above the Af temperature, in order to study the superelastic effect. In Figure 14, two superelastic tests at 115 °C and 130 °C, up to a maximum strain of 1.5%, are plotted. In both cases, the superelastic behavior exhibits a fully closed stress–strain cycle, and an important strain hardening is observed, as expected from a polycrystal. For the maximum strain of 2%, a slight residual deformation of about 0.1% was observed, indicating that further improvements are required to reach the superelastic recoverable strains observed in previous works on the powder metallurgy of these alloys [27]. Nevertheless, this is an outstanding result for a Cu–based SMA produced by LPBF. 

Then, the shape memory effect was tested by bending a small plate of the same sample LPBF + HIP–2 and TT–0. The 14 mm length, 5 mm width, and 1 mm thick sample was deformed by three–point bending tests up to a maximum surface strain (tension and compression with respect to the neutral plane) of about ± 2%; no extensometer was used in this case. These experiments were performed at room temperature, so in martensitic state, as it is presented in the left image of Figure 15, and a further heating with a hair dryer transformed the sample back to austenite, evidencing the recovery associated with the shape memory effect. The sequence is presented through several images in Figure 15, and recorded in a video illustrating the recovery process, which is shown in the Appendix A.

Many cycles of straining and recovery by heating were performed, and the sequence in Figure 15 evidences the shape memory effect, which exhibits a good shape recovery in all cycles. After some training, the recovery reaches about 95%. It is worth noting that this shape memory effect was not previously reported in Cu–Al–Ni SMAs produced by LPBF.

## 5. Conclusions

In the present work, the complete additive manufacturing (AM) processing route of Cu–13.5Al–4.0Ni (wt.%) shape memory alloys has been approached, from the powder production by gas atomization to the final processing through laser powder bed fusion (LPBF). The alloy was designed to exhibit the martensitic transformation above room temperature in order to test the presence of the shape memory effect by heating. The microstructure and the martensitic transformation behavior were comparatively studied along with the different processing steps: atomized powder, LPBF–processed samples with single and double melting, and a post–processing HIP followed by the thermal treatments for functionalization of the alloy. In the light of the presented experimental results and the corresponding analysis and discussion, the following conclusions can be drawn:Clean gas atomization is a primary key factor to obtain reliable powders with a minor deviation from the target chemical composition, particularly on the Al due to its extremely high influence on the martensitic transformation in these SMAs.The hot isostatic pressing (HIP) constitutes an excellent post–processing step to obtain the full compaction required for functional materials like the SMAs.After the LPBF processing, or after LPBF + HIP, some thermal treatments are required to functionalize the Cu–Al–Ni SMAs in order to restore the microstructure, avoiding the precipitation of the stable phases produced during the AM processing.The Cu–Al–Ni SMAs processed by LPBF and LPBF + HIP, thermally treated for functionalization, offer an excellent behavior regarding the thermoelastic martensitic transformation, exhibiting a low thermal hysteresis of about 16 °C.Reproducible and fully closed superelastic compression cycles up to 1.5% were obtained in the Cu–Al–Ni SMAs sample thermally treated after LPBF + HIP processing.A good shape memory effect by heating was obtained in Cu–Al–Ni SMAs processed by LPBF + HIP and thermally treated, paving the road for further studies.

The presented results validate the technique of LPBF for the additive manufacturing of Cu–Al–Ni SMAs, provided that the processing method be optimized. In addition, the composition of the alloy should be specifically designed for AM, in order to obtain a satisfactory match with the targeted transformation temperatures.

Finally, we may conclude that the way has been opened for the additive manufacturing of near–net–shape components of Cu–based SMAs. However, the design of specific alloys for AM, as well as the optimization of the processing and post–processing parameters to exploit the complete functional properties of these SMAs, still require further research.

## Figures and Tables

**Figure 1 materials-15-06284-f001:**
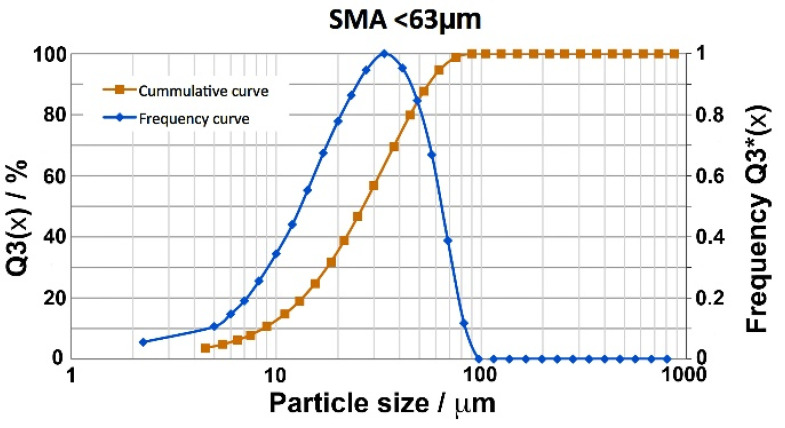
Particle size distribution: frequency and cumulative integral curves for the atomization process of the Cu–Al–Ni SMAs.

**Figure 2 materials-15-06284-f002:**
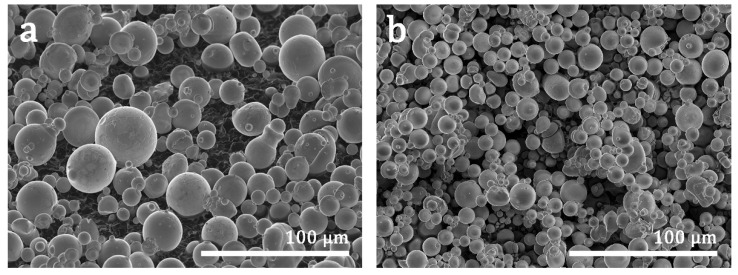
(**a**,**b**) SEM–SE images of the SMA powder showing different sizes of particles with a diameter below 63 µm. (**a**) Medium size particles 20–45 μm; (**b**) Small size particles 0–20 μm.

**Figure 3 materials-15-06284-f003:**
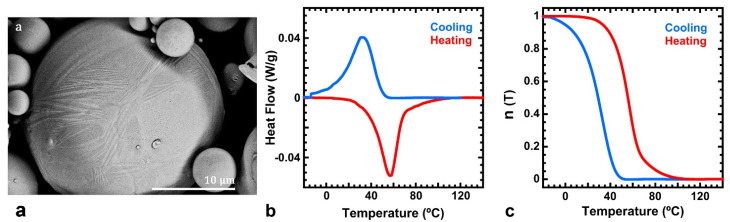
(**a**) SEM–BSE image of a powder grain. The martensite variants can be clearly seen on its surface in some of the bigger grains. (**b**) DSC results for the powder material. (**c**) Phase transformed fractions n(T), obtained by integrating the DSC results in (**b**), which are used to determine the MT temperatures.

**Figure 4 materials-15-06284-f004:**
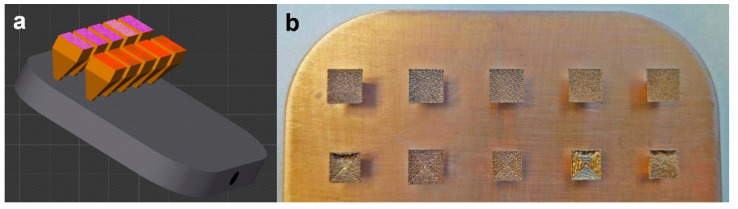
(**a**) Working table of the reduced built volume accessory. (**b**) An example of several samples built with different parameters and scanning strategies.

**Figure 5 materials-15-06284-f005:**
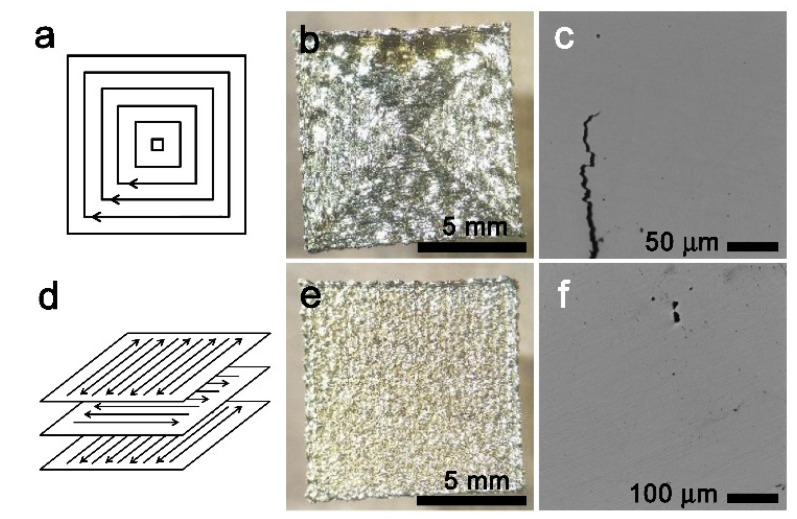
Scanning patterning strategies: (**a**) total fill scanning pattern (TF) and (**d**) meander scanning pattern (M). These methods give the pieces a final appearance shown in (**b**,**e**), respectively. (**c**,**f**) Optical micrographs of the corresponding samples.

**Figure 6 materials-15-06284-f006:**
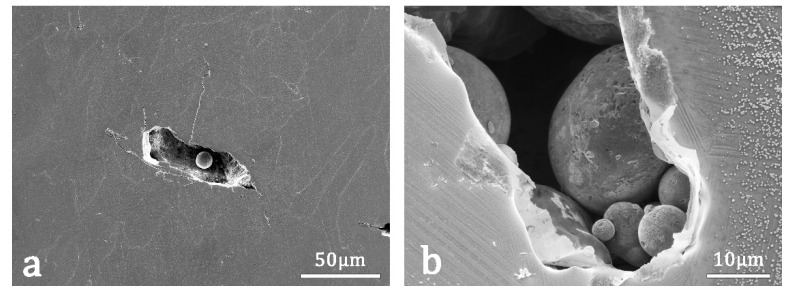
(**a**) SEM–SE image, taken in the SM–1 sample, of the porosity due to lack of fusion associated with a low local surface energy. (**b**) Another similar image in sample SM–1, showing not only the porosity but also the precipitates produced during LPBF processing.

**Figure 7 materials-15-06284-f007:**
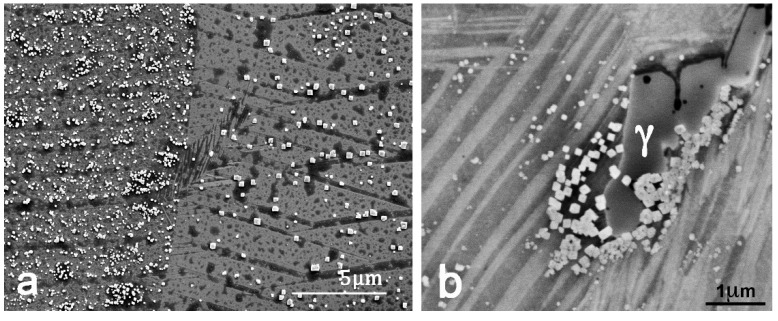
(**a**) SEM–SE image of as–LPBF DM–3 sample, with a distribution of small cuboidal precipitates (<<1 µm). (**b**) SEM–SE image of as–LPBF DM–4 sample showing the same cuboidal precipitates and a bigger γ precipitate. Martensites are clearly seen in both images.

**Figure 8 materials-15-06284-f008:**
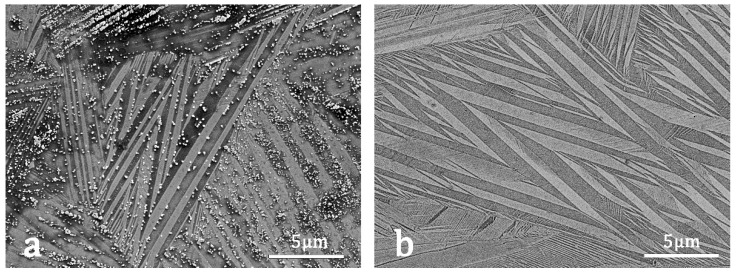
SEM images of the SM–1 treated samples. (**a**) SEM–SE after a slow quench at 100 °C (TT–100). The precipitates of the stable phase γ and out of stoichiometry α phase are observed. (**b**) SEM–BSE after a faster quenching in iced water and aging 24 h at 180 °C (TT–0); only the martensite variants are observed.

**Figure 9 materials-15-06284-f009:**
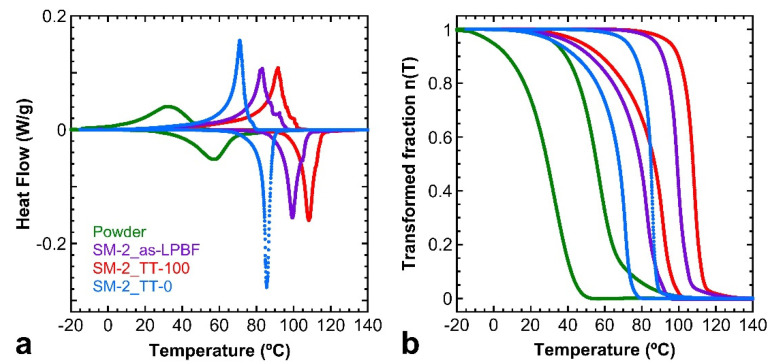
(**a**) DSC thermograms. (**b**) Transformed mass fraction curves. In both figures, different samples are compared: powder material (green curve), the as–LPBF sample SM–2 (violet curve), and after different thermal treatments, TT–100 (red curve) and TT–0 (blue curve). See text to correlate these curves with the different microstructural states.

**Figure 10 materials-15-06284-f010:**
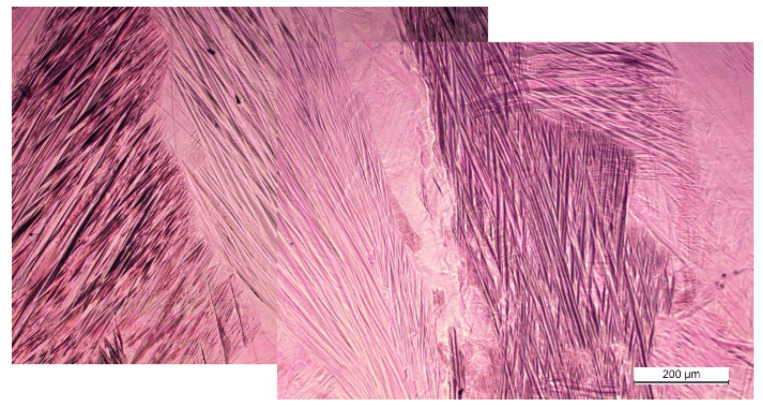
Optical microscopy by interferential contrast of the HIP–1 treated sample. The porosity is dramatically reduced and precipitates are not observed.

**Figure 11 materials-15-06284-f011:**
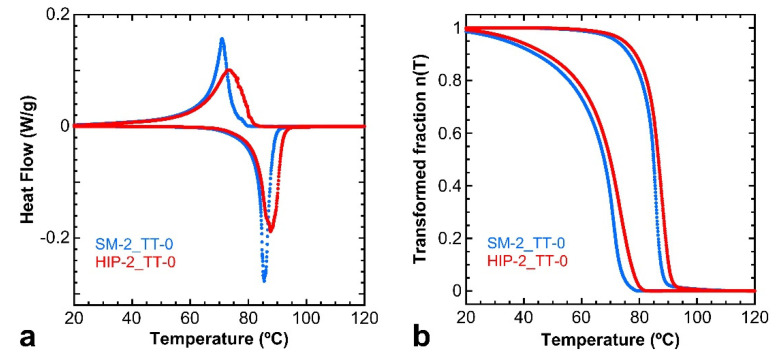
(**a**) DSC thermograms. (**b**) Transformed mass fraction curves. In both figures, the HIP–2 sample (red curves) and the SM–2 sample (blue curves) are compared after undergoing the TT–0 heat treatment.

**Figure 12 materials-15-06284-f012:**
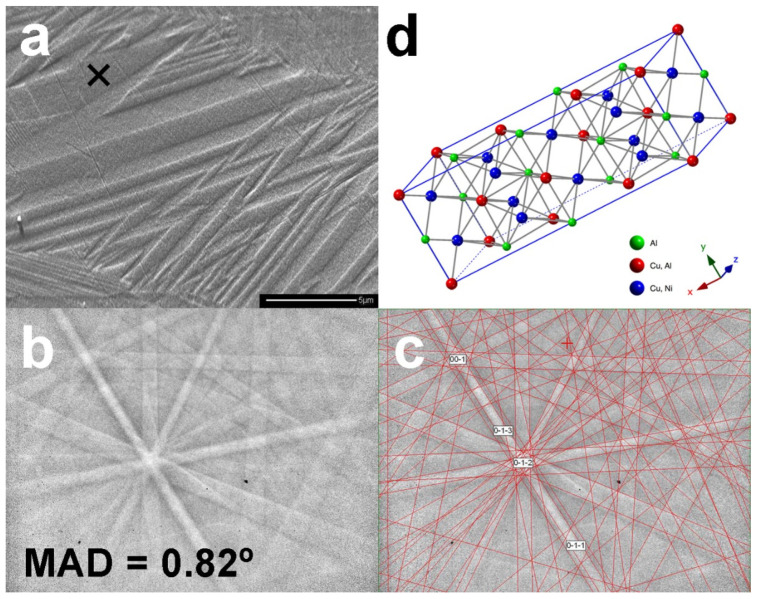
EBSD measurements on the LPBF–fabricated material (SM–2). (**a**) SME–BSE image of martensite variants. (**b**) EBSD pattern corresponding to the point marked with an x in (**a**). (**c**) Indexed EBSD pattern indicating the main Kikuchi lines. (**d**) Monoclinic $\beta_{3}^{\prime }$ martensite orientation of the pattern indexed in (**c**). The MAD parameter indicates the quality of the fitting.

**Figure 13 materials-15-06284-f013:**
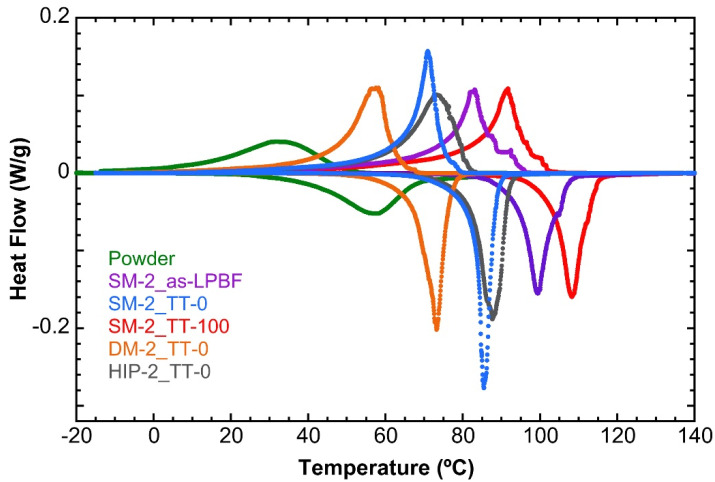
Summary of the obtained DSC results. Atomized powders (green curve) and as–LPBF single melt sample (violet curve). LPBF samples after slow quench TT–100 treatment (red curve) and after fast quench TT–0 treatment (blue curve). LPBF single melt plus HIP–treated TT–0 (dark grey curve) and LPBF double melt plus TT–0 treatment (orange curve). See the text for the explanation and discussion.

**Figure 14 materials-15-06284-f014:**
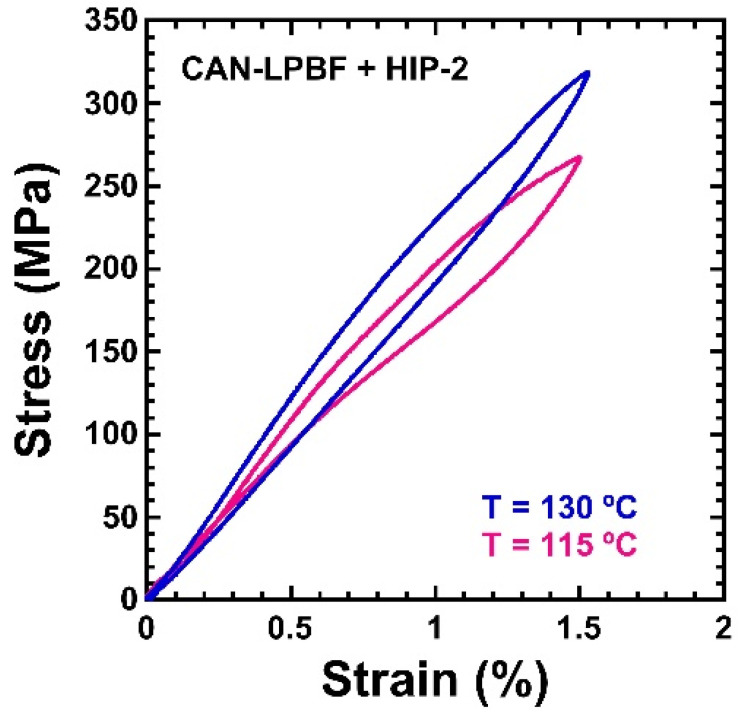
Compression superelastic cycles on the LPBF single melt sample after HIP–2 and TT–0 thermal treatment, up to a maximum strain of 1.5%, at 130 °C and 115 °C.

**Figure 15 materials-15-06284-f015:**
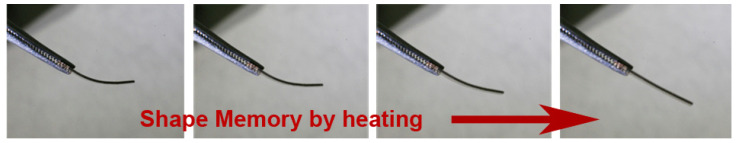
Images showing the shape memory effect obtained in the LPBF single melt sample after HIP and TT–0 thermal treatment. The sample, 14 mm long, was deformed at room temperature, so in martensite, and heated with a hair dryer to recover the shape in austenite.

**Table 1 materials-15-06284-t001:** Obtained weights and chemical composition, measured by ICP and LECO, for the different gas atomization of the same Cu–Al–Ni alloy.

Materials	Initial Weight (g)	<63 μm (g)	Cu (wt.%)	Al (wt.%)	Ni (wt.%)	O ppm	N ppm	C ppm	S ppm
Target	–	–	82.5	13.5	4.0	–	–	–	–
Atom 1	3000.5	2283.2	83.0	13.4	3.6	207	8	29	1
Atom 2	3000.5	2104.1	82.8	13.4	3.8	156	4	24	1

**Table 2 materials-15-06284-t002:** Parameters used for the different scan strategies in single melted (SM), double melted (DM), and samples with a further post–processing HIP. Codes: P**,** laser power; d_h_**,** hatch distance; v, scanning speed; E**,** energy per surface area; TF**,** total fill; M, meander.

Sample Codes	First Laser Scan					Second Laser Scan				
	P [W]	d_h_ [mm]	v [mm/s]	E [J/mm^2^]	Scanning Strategy	P [W]	d_h_ [mm]	v [mm/s]	E [J/mm^2^]	Scanning Strategy
SM–1	5	0.05	2500	0.04	TF	220	0.1	2000	1.1	M
SM–2	220	0.1	2000	1.1	M	5	0.05	2500	0.04	TF
DM–1	220	0.16	800	1.72	M	220	0.16	800	1.72	TF
DM–2	220	0.04	3200	1.72	M	220	0.04	3200	1.72	TF
DM–3	220	0.1	2000	1.1	TF	220	0.1	2000	1.1	TF
DM–4	220	0.04	3200	1.72	TF	220	0.04	3200	1.72	TF
HIP–1	250	0.04	800	7.81	M	5	0.05	2500	0.04	TF
HIP–2	250	0.04	1200	5.21	M	5	0.05	2500	0.04	TF

**Table 3 materials-15-06284-t003:** Martensitic transformation temperatures, thermal hysteresis, and transformation enthalpies (forward and reverse) for all the processing samples described in the present work and depicted in Figure 13.

CAN	°C					J/g
	Ms	Mf	As	Af	ΔT	ΔH
Powder	47.5	−8.3	25.9	91.6	26.4	7.5
SM–2 as–LPBF	92.8	30.2	81.3	115.4	19.7	8.3
SM–2 TT–0	76.4	23.5	65.2	90.7	17.2	8.4
SM–2 TT–100	99.9	35.9	87.3	118.8	20.8	8.0
DM–2 TT–0	69.5	12.8	49.6	84.2	17.9	8.5
HIP–2 TT–0	79.9	28.1	68.4	92.6	16.7	8.9

## Data Availability

The data presented in this study are available on request from the corresponding author.

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
