# Peer review of "Designing for Shape Memory in Additive Manufacturing of Cu–Al–Ni Shape Memory Alloy Processed by Laser Powder Bed Fusion"

_materials, 2022, doi:10.3390/ma15186284_

Round 1

Reviewer 1 Report

In this work, the Cu-Al-Ni shape memory alloy was prepared by additive manufacturing. The martensitic transformation and the related functional behaviors were analyzed. The authors performed detailed analyses on the experimental results. Some suggestions were listed below.

(1) In Fig. 9(b), the title of Y axis, i.e., heat flow, should be corrected.

(2) The indexation of Kikuchi pattern shown in Fig. 12 should be given. Otherwise, the MAD value makes no sense.

(3) Line 361-362, “according the proposed nomenclature for martensites”, it seems that “to” was missed in the sentence.

Author Response

Thanks for your comments contributing to improve the quality of our manuscript. We have considered all your comments and modified the text of the manuscript accordingly. In what follows, we give a detailed answer to your comments, indicating the modifications done in the manuscript.

(1) Concerning the point (1), the Fig. 9 has been corrected and changed.

(2) Concerning the point (2), we have modified the Fig. 12 in order to add the indexation of the EBSD pattern. This required rearrange the figure and the figure caption, as the corresponding text of the manuscript. Concerning this figure and according the request of the Reviewer 2, we have also add a short comment (in page 11) to indicate that this kind of study was done in different points with the same result.

(3) The sentence has been corrected.

Reviewer 2 Report

Reviewer : COMMENTS FOR THE AUTHOR:

In the present work, the complete methodology of AM processing of Cu-Al-Ni SMA is approached, from powders production to laser powder bed fusion fabrication, followed by the thermal and microstructural characterization. The final additive manufactured alloy exhibits an excellent thermal martensitic transformation as well as good shape memory behavior. The data of this paper is detailed. The paper in present format looks satisfied for the publication. But the following comments should be well addressed in revised version.

(1)  The abstract and conclusions should be restricted (brief) to new experimental and/or theoretical findings.

(2)  Some information is too long and redundant for the main work. For example, 2. Materials and methods, 2.1. Selected alloy and powder preparation. The author provided large redundant information for explaining the preparation and characteristics. Please delete some redundancy information in the manuscript and made a more focuses on the new concepts and new insights that were gained in the excellent thermal martensitic transformation as well as good shape memory behavior.

(3)  In AM process, powder is melted and is not sintered. Why did the authors compare the microstructures in different atomized powders? Once the powder remelts, the powder microstructure doesn't matter anymore.

(4) Fig. 12, the EBSD results for different samples should be added and compared. And give an more detailed explanation of excellent thermal martensitic transformation as well as good shape memory behavior.

Author Response

Thanks for your comments contributing to improve the quality of our manuscript. We have considered all your comments and modified the text of the manuscript accordingly. In what follows, we give a detailed answer to your comments, indicating the modifications done in the manuscript.

(1) Following your suggestions, we have suppressed a general sentence in the abstract (page 1) as well as a conclusion point that had a general character (in page 15).

(2) We have suppressed two sentences in point 2.1 (pages 2 & 3) that were too general and perhaps redundant.

(3) In the manuscript we have never talk about sintering and we don’t compare microstructures with different atomized powders. However, to avoid any misleading, we have added a short sentence (in page 3) indicating explicitly that all powders from the two atomizations were mixed for further processing.

(4) Concerning the results of EBSD, we have modified the Fig. 12 in order to include the indexation (as was also requested by Reviewer 1), and following your suggestion, we have also included a comment (in page 11) indicating that this kind of study was done in different points along the samples with the same result. We considered that adding most figures of the same results is redundant and space consuming without offering any new information. 

Reviewer 3 Report

The article «Designing for Shape Memory in Additive Manufacturing of Cu-Al-Ni Shape Memory Alloy Processed by Laser Powder Bed Fusion» is written on a current topic and can be published without a doubt.

Below I will write a few comments that, in my opinion, can improve the article:

- The introduction does not mention WAAM - which is quite popular in AM

- In the introduction, you need to write in more detail about  post-processing thermal treatment

- The Materials and Methods section is written in great detail - studies of the material received are presented. However, part of the bending tests is not described. please add

- When considering Figures 9 and 13, one can see the asymmetry of the transformation - it is necessary to explain what this may be due to (appearance of some kind of intermediate or new phase?)

- What was the degree of shape recovery with an induced strain of 2%? Full form restoration: 98-100% Why is the “±” sign used in the text?

- Have you tried to increase the deformation by bending, for example, up to 4-5%? I know that these alloys are quite brittle - when did the specimen fail in bending? If this experiment was carried out - please provide information

I really enjoyed reading your article! Good luck!

Author Response

Thanks for your comments contributing to improve the quality of our manuscript. We have considered all your comments and modified the text of the manuscript accordingly. In what follows, we give a detailed answer to your comments, indicating the modifications done in the manuscript.

(1) Following your suggestion in point 1, we have included a very short comment (in page 2) to cite the technique of WAAM, but in the manuscript we focused only in LPBF, which is the used technique. The other AM techniques are referenced for the interested readers.

(2) Following your suggestion in point 2, we have included in the last paragraph of the introduction (page 2), a new sentence concerning the need of a post-processing treatment, and we have also modified accordingly the surrounding sentences.

(3) We have included (in page 15) a more complete description of the bending tests. However, this is not included in the Materials and Methods section because this section is mainly focused on the description of the processing methodology and the thermal characterization. So we have considered that for the reader is better to have such description when the mechanical tests are presented.

(4) We agree with the reviewer. Indeed, the transformation peaks in Fig. 9 and Fig. 13 exhibit some asymmetry, and we have added a new sentence (in page 13) explaining the origin of the asymmetry observed in the low-temperature side, which is a general observation, as well as the asymmetry observed in the high-temperature side, appearing only in some samples.

(5) First, to comment that the ± sign is related to the tension and compression strains respect to the neutral plane of bending, and a short comment is added (in page 15) when describing the bending test. In what concerns the shape recovery, a recovery of 95% has been measured after training for a 2% strain. This is also indicated in a new sentence in the manuscript.

(6) In fact, the polycrystalline samples of Cu-Al-Ni SMA are very brittle because of the high elastic anisotropy generate a high stress concentration at he triple points of the grain boundaries. A maximum of 4% superelasticity was reported in the literature in samples produced by powder metallurgy (see Ref [27]). In our case, above the 2% of strain in compression, we start to observe some residual deformation of 0.1% during the superelastic tests, and a brittle fracture was obtained by cycling above 2.5% strain. We have not tested the samples by bending up to 4-5%, because we anticipate a premature fracture. This is the reason why we conclude that the optimization of the functional properties of samples produced by AM, still requires further works.
